# An Approximate Method for Predicting the Friction Factor of Viscoplastic Shear-Thinning Fluids in Non-Circular Channels of Regular Cross-Sections

**DOI:** 10.3390/polym14245337

**Published:** 2022-12-07

**Authors:** Mehmet Ayas, Jan Skočilas, Jan Štípek, Carlos Arce Gutiérrez, Rudolf Žitný, Tomáš Jirout

**Affiliations:** Department of Process Engineering, Czech Technical University in Prague, 16600 Prague, Czech Republic

**Keywords:** viscoplastic shear-thinning fluids, non-circular channels, laminar flow, Herschel–Bulkley model, friction factor

## Abstract

The objective of this study is to provide a straightforward generalized simple and quick method for the prediction of the friction factor for fully developed laminar flow of viscoplastic shear-thinning fluids in non-circular channels of regular cross-sections. The most frequently represented substances processed under these conditions are polymers in the processing and plastics industry. A generalized approximate method was proposed to express the relationship between the friction factor and the Reynolds number for the Herschel–Bulkley rheological model. This method uses the generalized Reynolds number for power-law fluids. Moreover, an additional simplified method for rapid engineering calculations was obtained as well. The suggested method was verified by comparing experimental data for concentric annulus found in the literature and results from simulations for concentric annulus, rectangular, square duct with a central cylindrical core and elliptical cross-sections. The results showed that the suggested methods enable us to estimate the friction factor with high accuracy for the investigated geometries.

## 1. Introduction

Viscoplastic fluids are an important class of non-Newtonian fluids and can be encountered in a variety of applications in the polymer industry [1,2,3,4]. The identifying mark of viscoplastic fluids is the presence of yield stress. Such fluids start to flow when the imposed shear stress is greater than the yield stress. On the other hand, when the applied shear stress is smaller than the yield stress, the material inside a duct behaves like an elastic plumb [5,6]. Polymer melts and biopolymers possess yield stress which ranks them among viscoplastic fluids, and their flow exhibits shear-thinning behavior [7,8,9,10,11].

The friction factor is one of the most frequently used non-dimensional design parameters in order to predict frictional pressure drop in channels or equipment which is necessary to calculate pump capacity [12], in material handling [13], designing heat exchangers [14], designing extrusion dies [15]. The generalized methods for calculation of hydraulic characteristics (friction factors) are usually restricted to zero yield stress liquids (power law models) and Bingham viscoplastic model [2,16]. Regarding the prediction of friction factor of fully developed viscoplastic shear-thinning fluids through channels, many of the works in literature are focused on eccentric-concentric annuli, parallel plates, circular duct, and open channels [16,17,18,19]. Few studies in the literature provide a generalized solution for friction-factor predictions in non-circular channels which rely on numerical methods [15].

The purpose of this article is to devise a straightforward expression that enables the calculation of pressure drop for the laminar flow of viscoplastic shear-thinning fluids (without thixotropy) through non-circular channels based on the Herschel–Bulkley rheological model. The method is based on the Rabinowitsch–Mooney equation and the friction factor–Reynolds number relationship is represented by using two geometrical parameters (a and b) suggested by Kozicki [20]. The suggested method is validated for concentric annulus, rectangular, square duct with a central cylindrical core, and elliptical cross-sections (Figure 1), by using numerical methods and experimental data found in the literature.

## 2. Materials and Methods

In the laminar regime, which is the most common regime in the polymer industry, the friction factor depends on the Reynolds number and cross-sectional geometry of the channel. In the case of Newtonian fluid flow, the friction factor–Reynolds number relation (*λ*–*Re*) is
(1)$$\lambda Re_{N}=C$$
where $Re_{N}$ is the Reynolds number for Newtonian flow, *C* is the constant depending on the geometry, and *λ* is the Fanning factor given by:(2)$$\lambda =\frac{\tau_{w}}{\frac{\rho \bar{u}{\;}^{2}}{2}}$$
where $\tau_{w}$ is the wall shear stress, ρ represents the density, and $\bar{u}\;$represents the mean velocity. Regardless of the rheological model of the fluid, $\tau_{w}$ is expressed by the following equation:(3)$$\tau_{w}=\frac{∆P}{L}\frac{D_{H}}{4}$$
where $D_{H}$ is the hydraulic diameter ($D_{H}$ = 4S/P), L represents the channel length, and ∆P the pressure drop along the channel. The power-law model is a frequently used rheological model to describe shear-thinning fluids, given by:(4)$$\tau =K{\dot{\gamma }}^{n}$$

In Equation (4), K is the coefficient of consistency, *n* is the flow index and $\dot{\gamma }$ is the shear rate (second invariant of the rate of deformation tensor). For the power-law model, fluid exhibits shear-thinning characteristics for 0<n<1, or shear-thickening (so-called dilatant) for *n* > 1. The unique case of *n* = 1 corresponds to the Newtonian case. The relationship between $\tau_{w}$ and $\bar{u}\;$was investigated extensively by Kozicki et al. [20] for the flow of purely viscous fluids in non-circular channels and various rheological models based on the Rabinowitsch–Mooney equation.

For the power-law fluids, the relationship of $\tau_{w}$ and $\bar{u}$ is described by Kozicki’s two-parameter model as follows:(5)$$\tau_{w}=K{\left[\left(b+\frac{a}{n}\right)\frac{8\bar{u}}{D_{h}}\right]}^{n}$$
where a and b are geometrical parameters. Their values for investigated geometries can be found in the publications of Kozicki et al. [20] and Sestak et al. [21]. For circular channel a = 0.25, b = 0.75, and for parallel plates a = 0.5, b = 1. The following relationships exist for the geometrical parameters a, b, and C:(6)$$a=\frac{C}{32}\frac{\bar{u}}{u_{max}}$$
(7)16(a+b)=C

The *λ–Re* relationship for power-law fluid can be obtained from Equations (2) and (5), giving:(8)$$\lambda Re_{G}=16$$

In Equation (8), $Re_{G}$ is generalized Reynolds given as:(9)$$Re_{G}=\frac{\rho \bar{u}{\;}^{2-n}D_{h}{}^{n}}{8^{n-1}K{\left(b+a/n\right)}^{n}}$$

Deplace and Leuilet [2] simplified Equation (8) by reducing two parameters, a and b, to one geometrical parameter—C, as follows:(10)$$v=\frac{b}{a}=\frac{48}{C}$$

As mentioned above, the Bingham model is one of the simplest two-parameter rheological models for viscoplastic fluids, described by the following equations:(11)$$\tau =\tau_{0}+\mu_{P}\dot{\gamma }\;\;\;\;\;\;\;\;\;\;\;\;\;\;\;\mathrm{If}\;\tau >\tau_{0}$$
$$\;\;\;\;\;\;\;\;\;\;\;\;\;\;\dot{\gamma }=0\;\;\;\;\;\;\;\;\;\;\;\;\;\;\;\mathrm{If}\;\tau \leq \tau_{0}$$

In Equation (13), $\mu_{P}$ is the plastic viscosity and $\tau_{0}$ is the yield stress. For the flow of Bingham fluids, the relationship between $\tau_{w}$ and $\bar{u}$ for non-circular channels is given by Kozicki’s equation [20] as follows:(12)$$\frac{8\bar{u}}{D_{h}}=\frac{\tau_{w}}{\mu_{p}}\left[\frac{1}{\left(a+b\right)}-\frac{\emptyset }{b}+\frac{a\emptyset^{\frac{b}{a}+1}}{b\left(a+b\right)}\right]$$

In Equation (12), the term ∅ is given by:(13)$$\emptyset =\frac{\tau_{0}}{\tau_{w}}$$

After rearranging, Equation (12) can be expressed alternatively as follows:(14)$$\frac{8\bar{u}}{D_{h}}=\frac{\tau_{w}}{\mu_{p}}\frac{\left(1-\emptyset \right)}{\left(a+b\right)}\left[1-\frac{a\emptyset }{b}-\frac{a\emptyset^{2}}{b}-\frac{a\emptyset^{3}(1-\emptyset^{\frac{b}{a}-2})}{b\left(1-\emptyset \right)}\right]$$

Another frequently used model to describe viscoplastic shear-thinning fluids is the Herschel–Bulkley model which involves three parameters:(15)$$\tau =\tau_{0}+K{\dot{\gamma }}^{n}\;\;\;\;\;\;\;\;\;\;\;\;\;\;\;\mathrm{If}\;\tau >\tau_{0}$$
$$\;\;\;\;\;\;\;\;\;\;\;\;\;\dot{\gamma }=0\;\;\;\;\;\;\;\;\;\;\;\;\;\;\;\mathrm{If}\;\tau \leq \tau_{0}$$

For the Herschel–Bulkley model, when using the Rabinowitsch–Mooney equation, the relationship of $\tau_{w}$ and $\bar{u}$ for the circular pipe is described as:(16)$$\frac{8\bar{u}}{D}={\left[\frac{\tau_{w}}{K}\right]}^{\frac{1}{n}}\frac{4n}{3n+1}{\left(1-\emptyset \right)}^{\frac{1}{n}}\left[1-\frac{\emptyset }{\left(2n+1\right)}-\frac{2n\emptyset^{2}}{\left(2n+1\right)\left(n+1\right)}-\frac{2n^{2}\emptyset^{3}}{\left(2n+1\right)\left(n+1\right)}\right]$$
and for parallel plate, the relationship of $\tau_{w}$ and $\bar{u}$ is:(17)$$\frac{8\bar{u}}{D_{h}}={\left[\frac{\tau_{w}}{K}\right]}^{\frac{1}{n}}\frac{2n}{2n+1}{\left(1-\emptyset \right)}^{\frac{1}{n}}\left[1-\frac{\emptyset }{\left(n+1\right)}-\frac{n\emptyset^{2}}{\left(n+1\right)}\right]$$

From Equations (14), (16), and (17), the relationship of $\tau_{w}$ and $\bar{u}$ for the non-circular channels in terms of geometrical parameters *a* and *b* can be approximately expressed for the Herschel–Bulkley model as follows:(18)$$\frac{8\bar{u}}{D_{h}}={\left[\frac{\tau_{w}}{K}\right]}^{\frac{1}{n}}\frac{{\left(1-\emptyset \right)}^{\frac{1}{n}}}{\left(b+\frac{a}{n}\right)}\left[1-\frac{\emptyset }{\left(v-1\right)n+1}-\frac{\left(v-1\right)n\emptyset^{2}}{\left[\left(v-1\right)n+1\right]\left[\left(v-2\right)n+1\right]}-\frac{\left(v-1\right)n^{2}\emptyset^{3}(1-\emptyset^{n\left(v-2\right)})}{\left[\left(v-1\right)n+1\right]\left[\left(v-2\right)n+1\right]\left(1-\emptyset^{n}\right)}\right]$$
where v = b/a. If n=1, Equation (18) reduces to Equation (14). If ∅=0, Equation (18) reduces to Equation (5). For circular cross-sections (v=3), Equation (18) becomes Equation (16) and for parallel plate (v=2), Equation (18) is equal to Equation (17).

Substituting Equation (18) to Equation (2), the *λ*–*Re* relationship for the fully developed, laminar, non-circular channels for the Herschel–Bulkley model can be expressed as:(19)$$\lambda Re_{G}=\frac{16}{\left(1-\emptyset \right)\theta^{n}}$$
and in Equation (19) the term *θ* is given by:(20)$$\theta =\left[1-\frac{\emptyset }{\left(v-1\right)n+1}-\frac{\left(v-1\right)n\emptyset^{2}}{\left[\left(v-1\right)n+1\right]\left[\left(v-2\right)n+1\right]}\left[1-\frac{n\emptyset (1-\emptyset^{v-2})}{\left(1-\emptyset \right)}\right]\right]$$

The term ∅ given in Equation (13) can be described in terms of $\tau_{0}$ and λ as follows:(21)$$\emptyset =\frac{\tau_{0}}{\frac{\lambda \rho \bar{u}{\;}^{2}}{2}}$$

Equation (18) is relatively complex; simpler expression can be obtained by eliminating its last term. Therefore, a simpler expression for the term θ given in Equation (20) can be written as follows:(22)$$\theta_{s}=\left[1-\frac{\emptyset }{\left(v-1\right)n+1}\left[1-\frac{\left(v-1\right)n\emptyset }{\left[\left(v-2\right)n+1\right]}\right]\right]$$

Hence, friction factor–Reynolds number expression can be stated as:(23)$$\lambda Re_{G}=\frac{16}{\left(1-\emptyset \right)\theta_{s}{}^{n}}$$

Calculation procedures for the estimation of the friction factor are given in Appendix A. In addition, the parameter v can be obtained from the method suggested by Delplace and Leuliet [2] given in Equation (10) for a simpler prediction of friction factor. In this study, the critical Reynolds number for the onset of turbulence is considered as 2000 for $Re_{G}$.

## 3. Validation

The validation of the suggested method is carried out by comparison with experimental data available in the literature, and by simulations. Fordham et al. [19] suggested a practical method in order to predict pressure drop in the concentric annulus and proposed a method validated experimentally using 0.5% aqueous solution of xanthan gum (*τ*_0_ = 1.59 Pa, K = 0.143 Pa.s^n^, *n* = 0.54). The comparison between predicted pressure drop values from Equation (19) and experimental data provided by Fordham et al. [19] for the concentric annulus is shown in Figure 2. The results indicate that the suggested method has a good relationship to experimental data. The maximum deviation has been found to be less than 10% and the average deviation was found to be 5%.

Ahmed [22] investigated the accuracy of hydraulic models by experimentally predicting the pressure loss for the isothermal laminar flow of viscoplastic shear-thinning fluids in concentric and eccentric annuli (in this study, only experimental results of the concentric annulus are considered). In experiments, xanthan gum (XCD, $\tau_{0}$ = 9.1 Pa, K = 1.01 Pa.s^n^, *n* = 0.48) and mixtures of xanthan gum and polyanionic cellulose (XCD-PAC, $\tau_{0}$ = 3.8 Pa, K = 2.98 Pa.s^n^, *n* = 0.4) were used as test fluids. Evaluated pressure losses by Equation (19) and its comparison with experimental results are presented in Figure 3.

It was found that the maximum difference between pressure loss obtained from Equation (19) and experimental results was less than 8%, and the average deviation was obtained as 6%.

In addition, the provided method has been verified by employing simulations using commercial software ANSYS Fluent 19.2. The validation was implemented by comparing $\lambda Re_{G}$ values acquired numerically and from Equation (19). Three-dimensional simulations have been carried out for steady-state, incompressible, isothermal, laminar flow of Herschel–Bulkley fluids in concentric annuli, square duct with a central cylindrical core rectangular, and elliptical cross-sectional geometries. The axial lengths of the channels were taken to be greater than 50D_h_ and half of the channels have been modeled due to the symmetry. Mesh sensitivity analysis was performed to obtain optimum mesh configuration and numbers of mesh elements for simulations. The procedure was carried out by comparing the effect of the number of mesh elements on obtained friction factor (λ) by simulations shown in Figure 4. Hence, four different numbers of mesh elements were tested, and 140,000–200,000 structural mesh elements were used in simulations (Figure 5). The deviation between points represents the mesh density effect. The optimum mesh represents the acceptable numerical error and computational time.

Especially fine mesh elements were preferred next to the wall and coarse elements were created in the middle sections [23]. The simulations were conducted for $Re_{G}<100$. Regarding the boundary conditions, inlet velocity and outlet pressure boundary conditions were applied at the inlet and outlet sections, respectively. Channel walls were chosen as stationary, with no slip on the wall. Symmetry boundary condition was imposed on symmetry planes. The SIMPLE algorithm was used for the velocity–pressure coupling. The second-order discretization scheme was adopted for pressure and momentum equations along with Green–Gauss node-based gradient option. Simulations were assumed to be converged when the residuals of continuity fell below 10^−8^.

Friction factor values were computed from the pressure drop values within the fully developed flow region of the channels. Calculated ∅ values from the result of simulations were between 0.04 and 0.62. The comparison between obtained $\lambda Re_{N}$ values using Equation (19) and simulations are displayed in Figure 6. From the acquired results, one can deduce that $\lambda Re_{N}$ values obtained by Equation (19) are slightly higher than those predicted by simulations. The deviations decrease with increasing values of *n* for all investigated geometries. The highest deviations were found for *n* = 0.3 which corresponds to the highest ∅ values. For rectangular channels, the suggested method shows a good agreement with the simulation results. The maximum deviation was found to be 7% and the average deviation was obtained as 5% between the two methods. For concentric annulus, the deviations between the suggested method and numerical results are less than 5% and the highest disagreement was found for *R_i_*/*R_o_* = 0.2. Regarding the elliptical channels, deviations increase with decreasing values of X/Y, and the maximum difference was found as 4%. For the square duct with a central cylindrical core, the maximum deviation was found to be less than 4%. Regarding the simplified method given in Equation (23), the difference between predictions between Equation (19) and Equation (23) were found to be less than 4% for the investigated rheological parameters and geometries. The average deviation was found to be between 3–4% for investigated geometries except for rectangular channels. The comparisons reveal that the suggested method provides a determination of the friction factor with high accuracy for Herschel–Bulkley fluids.

## 4. Conclusions

In this study, a fully developed, isothermal, laminar flow of viscoplastic shear-thinning fluids in non-circular channels was investigated for the Herschel–Bulkley model and by a correlation proposed for the estimation of the friction factor. The demand for fast and simple calculations is placed on designers of machines and equipment in the plastics industry. A simple equation or script that would be part of the design software for the prediction of friction losses during polymer flow would certainly be beneficial. A new method was obtained by using flow equations for slit and circular channels for Herschel–Bulkley fluids and the method suggested by Kozicki et al. [20] for Bingham fluids. Then, the friction factor–Reynolds number correlation is expressed using the generalized Reynolds number proposed by Kozicki et al. [20] which requires two geometrical parameters. The suggested method was verified for concentric annuli by utilizing experimental results reported in the literature and by using simulations for concentric annuli, rectangular and elliptical cross-sections. The results of the comparison revealed that the suggested method enables the estimation of friction factor with a deviation of less than 10%, which is the standard acceptable error for machine designing. From the simulation results, it was found that the accuracy of the suggested method decreases with decreasing values of *n*. However, the highest found deviation was 7% for rectangular channels. Furthermore, smaller deviations were found when other geometries were used. Consequently, the results of validations showed that the proposed method enables the estimation of the laminar flow friction factor for Herschel–Bulkley fluids in non-circular channels with high accuracy and can be helpful for plastic engineers.

## Figures and Tables

**Figure 1 polymers-14-05337-f001:**
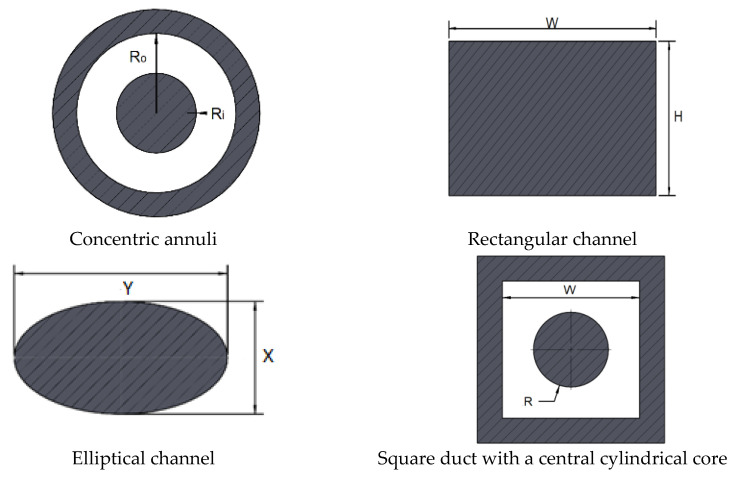
Investigated geometries.

**Figure 2 polymers-14-05337-f002:**
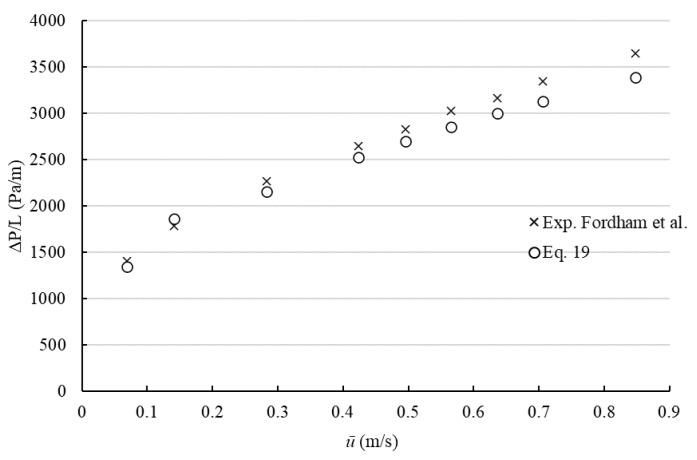
Comparison between Equation (16) and experimental data reported by Fordham et al. [19] for concentric annulus with $D_{i}$ = 0.04 m and $D_{o}$ = 0.05 m. (Adapted with permission from [19]. Copyright {1991} American Chemical Society).

**Figure 3 polymers-14-05337-f003:**
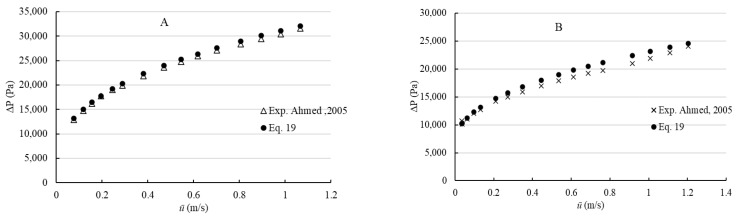
Comparison between Equation (19) and experimental data provided by Ahmed [22] for concentric annulus with $D_{i}$ = 0.0127 m and $D_{o}$ = 0.03505 m. (**A**) XCD-PAC, (**B**) XCD.

**Figure 4 polymers-14-05337-f004:**
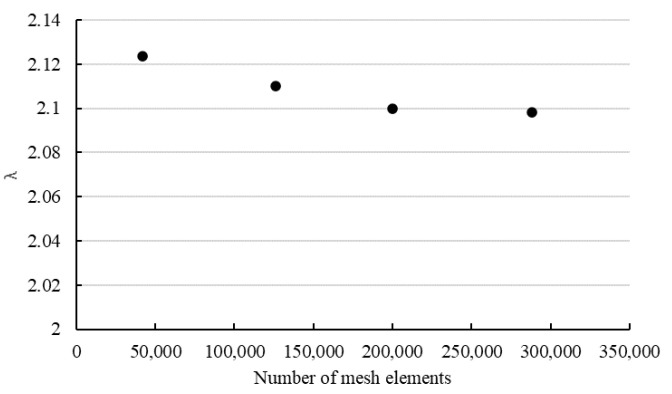
Mesh sensitivity analysis for concentric annulus of $R_{i}$/$R_{o}$ = 0.5, *τ*_0_ = 50 Pa, K = 5 Pa.s^n^, *n* = 0.5.

**Figure 5 polymers-14-05337-f005:**
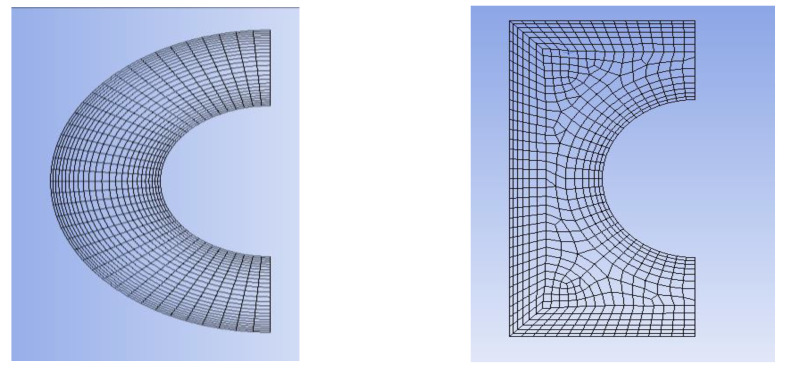
Generated meshes for simulations.

**Figure 6 polymers-14-05337-f006:**
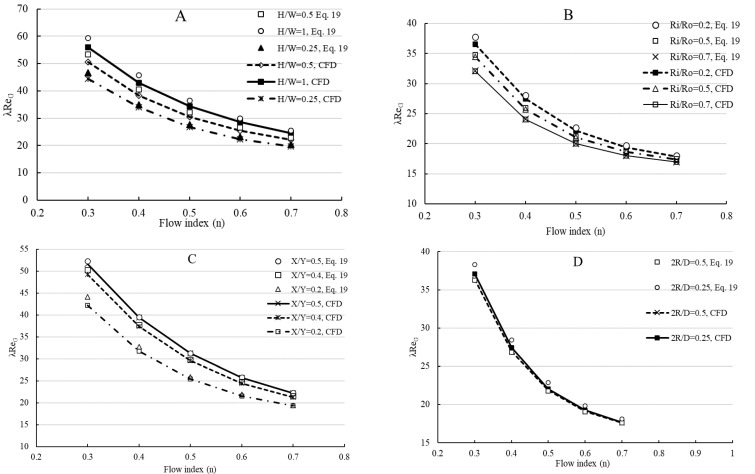
The comparison between evaluated $\lambda Re_{N}$ values from suggested method and simulations. (**A**) Rectangular channel, $\tau_{0}$ = 50 Pa, K = 5 Pa.s^n^, $\bar{u}$ = 0.1 m/s, W = 10 mm, 0.3 ≤ *n* ≤ 0.7; (**B**) Concentric annuli, $\tau_{0}$ = 50 Pa, K = 5 Pa.s^n^, $\bar{u}$ = 0.5 m/s, $D_{o}$ = 10 mm, 0.3 ≤ *n* ≤ 0.7; (**C**) Elliptical channel, $\tau_{0}$ = 50 Pa, K = 5 Pa.s^n^, $\bar{u}$ = 0.1 m/s, Y = 10 mm, 0.3 ≤ *n* ≤ 0.7; (**D**) Square duct with a central cylindrical core, $\tau_{0}$ = 50 Pa, K = 5 Pa.s^n^, $\bar{u}$ = 0.5 m/s, D = 10 mm, 0.3 ≤ *n* ≤ 0.7.

## Data Availability

Not applicable.

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
