# Peer review of "An Approximate Method for Predicting the Friction Factor of Viscoplastic Shear-Thinning Fluids in Non-Circular Channels of Regular Cross-Sections"

_polymers, 2022, doi:10.3390/polym14245337_

Round 1
Reviewer 1 Report
The authors worked on a problem titled "An approximate method for predicting the friction factor of viscoplastic shear-thinning fluids in non-circular channels of regular cross-sections." The approach to the problem is fine. Some similar studies performed by the authors themselves are cited in this paper. However, I needed to see major differences between those and the present study. I would like some clarity on this in the introduction. Moreover, some typo errors exist, such as Ref. [11].
Author Response
Point 1: The authors worked on a problem titled "An approximate method for predicting the friction factor of viscoplastic shear-thinning fluids in non-circular channels of regular cross-sections." The approach to the problem is fine. Some similar studies performed by the authors themselves are cited in this paper. However, I needed to see major differences between those and the present study. I would like some clarity on this in the introduction. Moreover, some typo errors exist, such as Ref. [11]
Response 1: Dear reviewer, thank you for your rewarding comments. Typo errors have been corrected. In the article, we had to mention about power-law case (Eq.8) and we wanted to add our works to mention about simplifation cases of Eq. 8. However, there was no any relation between suggested method in manuscript and our works mentioned in Ref. 22-23. Therefore, mentioned equations given in Ref. 22-23 (previous version of manuscript) have been removed and manuscript is more clear now.

Reviewer 2 Report
This manuscript investigated the friction factor of a shear-thinning fluid flowing in a non-circular channel of regular cross-section. The generalized Reynolds number for power-law fluids was used to represent the relationship between the friction coefficient and the Reynolds number in the H-B model. This research provided a simple method for calculating the friction factor in engineering applications. However, this work still needs to be improved further based on the following comments and suggestions.
1. Eq. 7 gave the relationship between the geometric parameters a, b and the constant C. It was easy to see that C=16 when a+b=1. So the λ-Re relation in the power law model given in Eq. 8 held only when a+b=1, but it was not stated in the manuscript. Please explain the range of application of Eq. 8. Justify if it also holds when a+b≠1.
2. The parameters in Table 1 showed that C=16 when a+b=1. Then the λ-Re relationship given in Eq. 25 was only applicable to some special shapes of channels. During the subsequent validation, it was evident that the channels used in the experiment did not satisfy the geometric condition of a+b=1. Similarly, please justify the applicability of the model to channels with geometrical parameters a+b≠1.
3. This manuscript mentioned several times in the validation section that the deviations of the model and the experiment are less than a certain value. Please indicate the specific values of these deviations, such as the maximum deviation and the average deviation.
4. Fig. 5 gave the sensitivity analysis of the Fluent simulation meshing. Please explain the meaning represented by the vertical coordinates of this picture and describe the analysis results reflected by this figure. The symbols of the vertical coordinates of this figure and λ in Eq. 2 are repeated, please distinguish them.
5. Magnetorheological fluids and shear-thickening fluids are typical non-Newtonian fluids. They satisfy the Bingham model and the power-law model, respectively. Magnetorheological shear-thickening polishing fluids have the properties of both magnetorheological fluids and shear-thickening fluids. They both have a wide range of applications in engineering. It is highly suggested to introduce these fluids in the introduction section of this paper. Related information can be found in the following references.
[1] Wei, M.H., Lin, K., Sun, L., Shear thickening fluids and their applications, Mater. Des., 2022, 216, 110570. https://doi.org/10.1016/j.matdes.2022.110570
[2] Qian, C., Tian, Y.B., Fan, Z.H., Sun Z.G., Ma Z., Investigation on rheological characteristics of magnetorheological shear thickening fluids mixed with micro CBN abrasive particles, Smart Mater. Struct., 2022, 31, 095004. https://doi.org/10.1088/1361-665X/ac7bbd
[3] Ma, Z.Q., Cao, J.G., Xu, X.H., Xu, J.H., A shear stress model of water-based magnetorheological polishing fluids, J. Intell. Mater. Syst. Struct., 2021, 33(1):160-169. https://doi.org/10.1177/1045389X211011660
[4] Fan, Z.H., Tian, Y.B., Zhou, Q., Shi, C., Enhanced magnetic abrasive finishing of Ti-6Al-4V using shear thickening fluids additives, Precis. Eng., 2020, 64, 300-306. https://doi.org/10.1016/j.precisioneng.2020.05.001
Author Response
Point 1: Eq. 7 gave the relationship between the geometric parameters a, b, and the constant C. It was easy to see that C=16 when a+b=1. So the λ-Re relation in the power law model given in Eq. 8 held only when a+b=1, but it was not stated in the manuscript. Please explain the range of application of Eq. 8. Justify if it also holds when a+b≠1.
Response 1: In Eq.8, the number 16 is an equation constant and doesn’t stand for a geometric parameter. Eq. 8 is a method suggested by Kozicki and geometrical parameters () are taken into account in the generalized Reynolds number given in Eq. 9.
|
The relationship between a, b and C are given in Eq. 7. If equation 8 is written for the Newtonian case along with Eq. 7 and the Generalized Reynolds number (Eq. 9), we can obtain the following expression
|
Where K is corresponding to the Newtonian viscosity. Consequently, as you see, The geometrical parameters are characterized by the Reynolds number according to the methodology.
Point 2 The parameters in Table 1 showed that C=16 when a+b=1. Then the λ-Re relationship given in Eq. 25 was only applicable to some special shapes of channels. During the subsequent validation, it was evident that the channels used in the experiment did not satisfy the geometric condition of a+b=1. Similarly, please justify the applicability of the model to channels with geometrical parameters a+b≠1.
Response 2
Since Eq. 25 is expressed by using the Generalized Reynolds number given in Eq. 8, the relevant answer is given in response 1.
Point 3
This manuscript mentioned several times in the validation section that the deviations between the model and the experiment are less than a certain value. Please indicate the specific values of these deviations, such as the maximum deviation and the average deviation.
Response 3
Mentioned maximum deviation and the average deviation are added and indicated in the manuscript.
Point 4
Fig. 5 gave the sensitivity analysis of the Fluent simulation meshing. Please explain the meaning represented by the vertical coordinates of this picture and describe the analysis results reflected by this figure. The symbols of the vertical coordinates of this figure and λ in Eq. 2 are repeated, please distinguish them.
Response 4
Figure 5 changed as figure 4. The description of the mesh sensitivity section is modified as requested and λ is specified in the manuscript.
“Mesh sensitivity analysis was performed to obtain optimum mesh configuration and numbers of mesh elements for simulations. The procedure was carried out by comparing the effect of the number of mesh elements on obtained friction factor (λ) by simulations shown in figure 4. Hence, four different numbers of mesh elements were tested, and 140000-200000 structural mesh elements were used in simulations (figure 5). The deviation between points represents the mesh density effect. Optimum mesh represents the acceptable numerical error and computational time. ”
Point 5
Magnetorheological fluids and shear-thickening fluids are typical non-Newtonian fluids. They satisfy the Bingham model and the power-law model, respectively. Magnetorheological shear-thickening polishing fluids have the properties of both magnetorheological fluids and shear-thickening fluids. They both have a wide range of applications in engineering. It is highly suggested to introduce these fluids in the introduction section of this paper. Related information can be found in the following references
Response 5
Based on your suggestion, one of the recommended articles is mentioned in the introduction section added in reference.
